# PDRN, a Bioactive Natural Compound, Ameliorates Imiquimod-Induced Psoriasis through NF-κB Pathway Inhibition and Wnt/β-Catenin Signaling Modulation

**DOI:** 10.3390/ijms21041215

**Published:** 2020-02-12

**Authors:** Natasha Irrera, Alessandra Bitto, Mario Vaccaro, Federica Mannino, Violetta Squadrito, Giovanni Pallio, Vincenzo Arcoraci, Letteria Minutoli, Antonio Ieni, Maria Lentini, Domenica Altavilla, Francesco Squadrito

**Affiliations:** 1Department of Clinical and Experimental Medicine, University of Messina, c/o AOU Policlinico G. Martino, Via C. Valeria Gazzi, 98125 Messina, Italy; nirrera@unime.it (N.I.); abitto@unime.it (A.B.); vaccaro@unime.it (M.V.); fmannino@unime.it (F.M.); violettasquadrito@gmail.com (V.S.); gpallio@unime.it (G.P.); varcoraci@unime.it (V.A.); lminutoli@unime.it (L.M.); 2Department of Human Pathology in Adult and Developmental Age “Gaetano Barresi”, University of Messina, c/o AOU Policlinico G. Martino, Via C. Valeria Gazzi, 98125 Messina, Italy; aieni@unime.it (A.I.); lentini@unime.it (M.L.); 3Department of Biomedical and Dental Sciences and Morphological and Functional Sciences, University of Messina, c/o AOU Policlinico G. Martino, Via C. Valeria Gazzi, 98125 Messina, Italy; daltavilla@unime.it

**Keywords:** adenosine A2A receptor, psoriasis, NF-κB, Wnt/β-catenin pathway

## Abstract

Nuclear factor-κB (NF-κB) plays a central role in psoriasis and canonical Wnt/β-catenin pathway blunts the immune-mediated inflammatory cascade in psoriasis. Adenosine A2A receptor activation blocks NF-κB and boosts the Wnt/β-catenin signaling. PDRN (Polydeoxyribonucleotide) is a biologic agonist of the A2A receptor and its effects were studied in an experimental model of psoriasis. Psoriasis-like lesions were induced by a daily application of imiquimod (IMQ) on the shaved back skin of mice for 7 days. Animals were randomly assigned to the following groups: Sham psoriasis challenged with Vaseline; IMQ animals challenged with imiquimod; and IMQ animals treated with PDRN (8 mg/kg/ip). An additional arm of IMQ animals was treated with PDRN plus istradefylline (KW6002; 25 mg/kg/ip) as an A2A antagonist. PDRN restored a normal skin architecture, whereas istradefylline abrogated PDRN positive effects, thus pointing out the mechanistic role of the A2A receptor. PDRN decreased pro-inflammatory cytokines, prompted Wnt signaling, reduced IL-2 and increased IL-10. PDRN also reverted the LPS repressed Wnt-1/β-catenin in human keratinocytes and these effects were abolished by ZM241385, an A2A receptor antagonist. Finally, PDRN reduced CD3+ cells in superficial psoriatic dermis. PDRN anti-psoriasis potential may be linked to a “dual mode” of action: NF-κB inhibition and Wnt/β-catenin stimulation.

## 1. Introduction

Psoriasis is an inflammatory skin disease with an important impact on patient quality of life. Plaques psoriasis is the most common type of the disease and presents red, well delineated, and silver plaques mainly localized in the umbilical and lumbosacral area as well as in the elbows, knees, and scalp. From a histological point of view, psoriasis plaques show a high proliferation rate and are characterized by immature keratinocytes and by an incomplete development of cornification, causing a marked thickness in epidermis with the occurrence of elongated rete ridges. The dermis is also characterized by the infiltration of an exaggerated number of macrophages, dendritic cells and T-cells. Furthermore, psoriasis is frequently associated with an uncontrolled inflammation of the musculoskeletal system; the main interested sites being the joints (arthritis), the tendons and ligaments of the bone (enthesitis), and the bone of the peripheral and assail skeleton. Indeed, it has been shown that in experimental models of psoriasis an impaired activity of T-cells represents a key fingerprint of the disease and an imbalance between immune system, inflammatory response and skin turnover is responsible for the appearance of psoriatic lesions [1]. Inflammation represents one of the most important hallmarks of psoriasis and pro-inflammatory cytokines deriving from T cells activation, such as Tumor Necrosis Factor alpha (TNF-α) and several Interleukins (IL), participate to the appearance of some psoriatic features such as epidermal hyperplasia, acanthosis, and hyperparakeratosis [2].

Methotrexate, one of the main prescribed drugs for the treatment of psoriasis, acts, at least in part, by augmenting the endogenous levels of adenosine, thus unmasking a role for this mediator in the inflammatory reaction that occurs under this pathological condition. Indeed, adenosine is an endogenous regulator of inflammatory processes, mediating the resolution of inflammation [3]. Four types of adenosine receptors have been identified and indicated as A1, A2A, A2B, and A3.

These receptors differ in affinity for adenosine and have a different level of expression in the several cell types. It has been proposed that targeting adenosine receptors, specifically the A2A subtype, might be considered a good strategy for the treatment of skin diseases characterized by inflammation [4,5].

Indeed, A2A receptor stimulation produces anti-inflammatory effects by inhibiting Nuclear factor κB (NF-κB) translocation to the nucleus and consequently suppressing TNF-α expression [6,7]. This could be relevant for the treatment of psoriasis.

Besides the NF-κB triggered inflammatory cascade, the Wnt/β-catenin signaling has also been suggested to positively modulate the inflammatory reaction of psoriasis [8]. Interestingly, Wnt signaling can be activated by the A2A receptor through an increase of the intracellular cAMP, which is a well-known intracellular second messenger that boosts Wnt signaling [9]. This molecular pathway might represent another rational target for an anti-psoriasis agent.

PDRN, polydeoxyribonucleotide, is a biologic compound that engages the A2A receptor and exerts a significant anti-inflammatory effect [10]. Moreover, PDRN is particularly effective on skin, improving repair and remodeling processes [11], and it is considered as a safety compound, lacking side effects compared to other anti-inflammatory drugs [12].

Therefore, the aim of this study was to investigate the efficacy of PDRN in an experimental model of psoriasis-like dermatitis.

## 2. Results

### 2.1. PDRN Treatment Ameliorates Skin Alterations Following IMQ Administration

Shaved back skin of mice challenged with Vaseline showed a normal appearance during all the experimental procedure, with no phenotypical or histological alterations (Figure 1A). IMQ application caused the occurrence of erythema and scales following three days of imiquimod challenge (Figure 1B). At day 7, consistent with clinical scores, histological analysis revealed that IMQ caused skin alterations, typical of psoriatic human skin: epidermal hyperplasia, acanthosis, and inflammation (Figure 1F). PDRN treatment ameliorated skin structure both macroscopically and microscopically (Figure 1C,G, respectively). In particular, PDRN administration reduced squamous lesions, erythema, epidermal thickness, and acanthosis, trying to restore a normal skin architecture (Figure 1C,G). Istradefylline, an A2A antagonist, abrogated the positive effects of PDRN in psoriatic skin (Figure 1D,H).

### 2.2. PDRN Reduces both pNF-κB and Pro-Inflammatory Cytokines Expression

A previous experiment has already demonstrated that the expression of the phosphorylated transcriptional factor NF-κB and consequently of the pro-inflammatory cytokine TNF-α increases following IMQ administration because of an inflammatory process triggering [13]. In this experimental setting too, mice challenged with IMQ showed an enhanced expression of both pNF-κB and TNF-α. IMQ mice treated with PDRN showed a marked reduction of pNF-κB and TNF-α expression compared to untreated psoriatic animals (Figure 2A,B).

Skin challenged with IMQ alone showed an increased mRNA expression of other pro-inflammatory cytokines, such as IL-6 and IL-12 compared to Sham mice. PDRN administration caused a marked decrease of both IL-6 and IL-12 (Figure 3), likely because of pNF-κB inhibition occurring upon adenosine receptor stimulation.

### 2.3. PDRN Anti-Inflammatory Activity Occurs through Wnt/β-Catenin Pathway Activation

Skin of IMQ mice showed a reduced expression of both Wnt-1 and β-catenin (Figure 4), thus demonstrating that IMQ caused canonical Wnt/β-catenin signaling impairment. Canonical Wnt/β-catenin pathway activation may modulate cytokines expression in particular IL-2 and IL-10 expression which are a pro- and anti- inflammatory cytokines, respectively. In agreement with this evidence, IMQ challenge caused a significant increase of IL-2 and a decrease of IL-10 mRNA expression.

PDRN treatment also boosted the activation of Wnt-1 and β-catenin in psoriatic animals. Moreover, PDRN reduced IL-2 expression and increased IL-10 mRNA expression, as a consequence of Wnt/β-catenin pathway activation (Figure 4).

An inflammatory in vitro model was reproduced to confirm the anti-inflammatory effect of PDRN by treating human keratinocytes with LPS for 24 h. The stimulation with LPS reduced the expression of both Wnt-1 and β-catenin compared to control cells. PDRN treatment significantly increased both molecules involved in Wnt/β-catenin signaling (Figure 5), thus confirming the results obtained in the in vivo model. These results were reverted following the use of the A2A receptor antagonist ZM241385.

Moreover, HaCaT cells were stimulated with TNF-α for 6 h and then treated with PDRN until 24 h to evaluate whether PDRN might modulate the inflammatory response induced by TNF-α. TNF-α incubation produced a marked increase of IL-6 expression, whereas as expected, PDRN significantly reduced the mRNA expression following 24 h of treatment (Figure 5).

### 2.4. PDRN Reduces CD3+ Cells

Immunohistochemical evaluation of representative skin samples revealed that skin collected from Sham mice showed a small amount of CD3+ cells (Figure 6A), whereas an increased number of CD3+ cells was observed in IMQ animals mainly in the dermo-epidermal junctions and in superficial dermis especially around hair follicles (Figure 6B). PDRN treatment reduced CD3+ cells while immunohistochemical images showed an intermediate number of CD3 elements both in junctions and in superficial dermis (Figure 6C).

### 2.5. PDRN Modulates Wnt-1/β-Catenin Signaling in Human Keratinocytes

Human keratinocytes were used to demonstrate PDRN effects on Wnt/β-catenin signaling. PDRN treatment significantly increased both Wnt1 and β-catenin mRNA expression compared to untreated keratinocytes.

CGS21680, a specific A2A receptor agonist, also augmented mRNA expression of Wnt-1 and β-catenin in human keratinocytes, thus confirming the A2A receptor involvement in Wnt/β-catenin pathway. The use of the A2A antagonist ZM241385 abrogated PDRN effects on Wnt-1 and β-catenin, thus confirming PDRN mode of action (Figure 7).

## 3. Discussion

Inflammation plays an important role in psoriasis and represents one of the most relevant features of the disease, worsening the prognosis [14]. Hyper-activation of cutaneous and immune cells is responsible for epidermal growth and triggers the release of pro-inflammatory cytokines.

Among the several pro-inflammatory cytokines involved in psoriasis, TNF-α, IL-6, and IL-12 are usually increased in psoriatic patients, creating a negative loop that exacerbates the inflammatory condition [15]. Preclinical findings point out that external stimuli, as in the so called Koebner phenomenon, trigger cutaneous damage that in turn leads to the release of antimicrobial peptides by keratinocytes. These signals, interacting with RNA o DNA molecules, causes the production of complexes able to trigger plasmacytoid dendritic cells (pDC) via toll like receptor (TLR) 7 and TLR9 intracellular signaling cascade. pDCs release type I interferons (IFNs), emphasizing the recruitment of myeloid dendritic cells and T-cells. Myeloid DCs in turn cause the production of IL-12 and IL-23. Both cytokines boost the activation and priming of helper T (TH) cells to differentiate towards a TH1 and TH17 phenotype, respectively. Primed TH1 release IFN-γ and tumor necrosis factor alpha (TNF-α), while TH17 cells release IL-17 and IL-22. These pro-inflammatory signals cause expansion of keratinocytes and further amplify cutaneous inflammation yielding to the formation of the characteristic psoriasis plaques. This impaired immunological scenario is confirmed by the marked and sustained therapeutic efficacy of the several novel biological drugs such as monoclonal antibodies against the p40 subunit shared by IL-12 and IL-23 (ustekinumab) and IL-17/IL-17 receptor (i.e., secukinumab, ixekizumab). These new drugs together with the inhibitor of phosphodiesterase 4 (PDE-4) apremilast, are today recognized as benchmarks in the management and therapy of moderate to severe psoriasis. Interestingly, apremilast blocks the intracellular cascade of molecular events that concur to the production and release of several pro-inflammatory cytokines, mainly IL-17, and therefore might work to correct the immunological imbalance of psoriasis [15].

In the intracellular cascade of events, nuclear factor κB plays a central role. This is a nuclear transcription factor that modulates the expression of the genes involved in the inflammatory processes. NF-κB is usually in its inactive state in the cytosol and associated with I-κB. When I-κBα is phosphorylated, NF-κB separates from I-κBα and translocates into the nucleus, thus activating the transcription of the genes encoding pro-inflammatory cytokines.

Under the present experimental design, IMQ application activated the transcriptional factor NF-κB and consequently promoted the expression of pro-inflammatory cytokines such as TNF-α, IL-6, and IL-12. In this context, IMQ caused characteristic histological skin alterations: more specifically epidermal hyperplasia, hyperparakeratosis and inflammatory infiltrate accumulation were observed.

The adenosine A2A receptor activation determines an important and useful anti-inflammatory effect, for this reason PDRN, thanks to its activity on adenosine receptors, may be considered as a new approach in pharmacotherapy. DNA nucleases located in plasma or in tissues are involved in the degradation of PDRN, thus forming oligo and mononucleotides. This is one of the main processes that characterize PDRN mechanism of action: the production of nucleosides and nucleotides following PDRN degradation mediates the binding to the A2A receptor [12]. Moreover, our previous immunofluorescence experiments on MC3T3-E1 cell line showed us a positive expression of A2A receptors following PDRN treatment (data not shown). PDRN anti-inflammatory activity has been highlighted not only in several experimental models [10,16,17] but also in patients. The present results confirmed the marked anti-inflammatory potential of this biologic compound: PDRN reduced NF-κB expression following IMQ administration and also tempered TNF-α, IL-6, and IL-12 expression. These results lead us to hypothesize that PDRN therapeutic effect might be ascribed, at least in part, to a reduction of immuno-inflammatory mediators. The decrease of inflammatory infiltrate and the reduction of epidermal hyperplasia and parakeratosis following PDRN treatment supported the molecular data.

Recently, a role of the canonical Wnt/β-catenin pathway has been described in psoriasis: more specifically it has been pointed out that Wnt signaling may be involved in the modulation of immune/inflammatory cascade. Interestingly, Wnt/β-catenin pathway is involved in a plethora of events, in particular during embryogenesis, stimulating developmental processes [18]. The activation of Wnt/β-catenin pathway requires the binding of Wnt proteins to Frizzled (Fz) receptors and the presence of the co-receptors LRP (low-density lipoprotein receptor-related proteins) 5 and 6. β-catenin is so activated and translocates into the nucleus, thus promoting gene transcription [19]. This mechanism falls into the “so called” canonical pathway, but alternatively, Wnt signaling may proceed through the non-canonical pathway involving Fz receptors, but independently from β-catenin activation [20].

Canonical Wnt/β-catenin pathway activation modulates immune-inflammatory mediators, such as IL-2 and IL-10 expression. In this experimental model, IMQ administration diminished Wnt1 and consequently β-catenin mRNA expression. Moreover, IMQ caused a significant increase and a marked decrease of IL-2 and IL-10 mRNA expression, respectively, likely as a consequence of Wnt1 and β-catenin pathway down-regulation.

Therefore “reawakening” the Wnt/β-catenin signaling may be an innovative strategy for the treatment of psoriasis. However, it remains to identify an effective tool that accomplishes this task. Targeting the adenosine A2A receptor offers an interesting potential. Indeed, canonical Wnt/β-catenin signaling may be activated by A2A receptor stimulation because cyclic AMP (adenosine monophosphate) accumulation increases Wnt/β-catenin expression [21]. In agreement with this experimental thesis, PDRN treatment enhanced both Wnt1 and β-catenin mRNA expression, thus restoring and boosting canonical Wnt/β-catenin signaling. These results were also confirmed in an in vitro setting: PDRN enhanced both Wnt-1 and β-catenin in human keratinocytes following 4 h of incubation. In contrast, this effect was abolished by ZM241385, an antagonist of the A2A receptor, thus demonstrating that the Wnt/β-catenin activation observed in mice occurred through the adenosine receptor stimulation. The treatment with CGS21680, a specific A2A receptor, further confirms the involvement of A2A receptor in Wnt/β-catenin pathway. In fact, this specific A2A receptor agonist also increased mRNA expression of both Wnt-1 and β-catenin in human keratinocytes.

In addition, TNF-α and LPS were used as stimuli to induce a psoriasis-like in vitro model. PDRN treatment significantly reduced the mRNA expression of IL-6 following stimulation of keratinocytes with TNF-α, confirming the anti-inflammatory effect of PDRN, as previously described [10,12]. Schön MP et al. have described a possible interaction of IMQ, used in the in vivo experimental paradigm, with the adenosine receptors A1 and A2 [22]. IMQ seems to be an adenosine receptor antagonist and positively stimulates the inflammatory response. However, the result obtained in human keratinocytes stimulated with TNF-α demonstrated that PDRN has an important potential as an anti-inflammatory drug through the modulation of the adenosine receptor pathway and independently of the imiquimod antagonism at the receptor level.

Wnt/β-catenin pathway was studied following LPS incubation: LPS reduced the expression of Wnt-1 and β-catenin in stimulated keratinocytes whereas PDRN significantly increased the mRNA expression of both, thus confirming the data obtained in psoriatic skin and in normal keratinocytes.

As a read-out of this triggering, PDRN blunted IL-2 and increased IL-10 mRNA expression in treated psoriatic mice compared to IMQ animals.

Psoriasis is a complex immune-mediated disease and, in this specific immunological setting, T cells activation plays a fundamental role [23]. IL-2 may be also produced by activated T cells promoting a feedback loop on T lymphocytes which release more IL-2 [24]. PDRN treatment significantly decreased CD3+ cells and caused IL-2 mRNA expression reduction, thus emphasizing a PDRN immunomodulatory activity, likely arising from its predominant anti-inflammatory effect.

The present experimental data suggest that PDRN anti-psoriasis potential may be linked to a “dual mode” of action: a NF-κB inhibitory activity that is combined with a stimulation of canonical Wnt/β-catenin pathway. Thus, the A2A receptor orchestrates a crosstalk between the two intracellular signaling cascades that results in positive modulation of both the inflammatory and immune reactions. PDRN is a biologic A2A agonist already in the market, even if, at the present, with different therapeutic claims: therefore, this study has a strong translational impact. However, the anti-psoriatic effect deserves additional studies and needs to be confirmed in a clinical setting. In addition, it is worthy of interest to stress another high translational potential linked to the PDRN pharmacology [12]. Indeed, there is a very low adherence of psoriatic patients to the systemic therapy that leads to the discontinuation of the treatment and to a worsening of the symptoms. This has prompted the interest in the use of topical treatment that leads to a higher patient compliance. In fact, topical drug therapy represents nowadays the cornerstone in the management of mild to moderate psoriasis. The advantages brought about by direct skin delivery of drugs are well recognized: it gives a direct target on the diseased skin that may reduce the systemic adverse events and it may have, in addition, a positive psychological impact. Interestingly, PDRN has shown a good pharmacokinetic and chemical profile that anticipates the feasibility of a topical formulation. Preliminary results from our laboratory (data on file) suggest that a PDRN gel formulation is bioequivalent to systemic PDRN and exerts the same protective effect in the imiquimod model of psoriasis.

Indeed, PDRN may share with apremilast, a benchmark drug for the treatment of psoriasis, an overlapping mode of action, at least on the intracellular signaling events that lead to the downregulation of the inflammatory cascade that plays a fundamental role in the pathogenesis of psoriasis. Activation of the A2A receptor induces an increase in the intracellular cyclic AMP as apremilast does through phosphodiesterase inhibition. The augmentation in cyclic AMP synergizes with other molecular signaling that cooperate to reduce the translocation of NF-κB to the nucleus and to turn off genes codifying for the master cytokines involved in the pathogenesis of psoriasis. However, stimulation by PDRN of this specific subtype of adenosine receptor has, as we have showed in the present paper, further effects on another signaling cascade that concur to the blunting of the exaggerated inflammatory reaction: the Wnt/β-catenin signaling. The modulation of this intracellular molecular pathway may cause a more robust and sustained suppression of the keratinocyte inflammatory phenotype, in turn limiting keratinocyte proliferation and plaque formation. However, at the moment, we do not know whether PDRN exerts a greater therapeutic activity when compared to apremilast, and future experimental work should be carried out to investigate this point. Finally, it should be pointed out that PDRN has a good safety profile [12]. Clinical trial, as well as postmarketing and pharmacovigilance studies [12] have confirmed that the drug is safe and well tolerated. Moreover, PDRN metabolism does not occur in the liver and this feature avoids the occurrence, during treatment with PDRN, of interaction with other drugs that may be concomitantly administered.

## 4. Materials and Methods

### 4.1. Animals, Experimental Model and Treatment

Balb/c mice (25–30 g; Charles River, Calco, Italy) were used in this study. During the experiment, mice were housed in plastic cages under controlled environmental conditions (12 h light–dark cycle, 24 °C) in the Animal Facility of the Department of Clinical and Experimental Medicine. Animals were provided with standard food and water ad libitum and all the experiments were performed in compliance with the standards for care and use of animals as stated in the Directive 2010/63/EU, and the ARRIVE guidelines [25]; the procedures were evaluated and approved by the Ethics Committee of the University of Messina (OPBA, #820/2016, 02/09/2016).

Psoriasis-like lesions were induced by a topical daily application of imiquimod cream (IMQ; 62.5 mg/day for 7 days) on the shaved back skin of mice (*n* = 30) for 7 consecutive days. Animals were randomly assigned to the following groups: sham psoriasis, which were challenged with Vaseline cream; IMQ animals, only challenged with imiquimod cream and IMQ animals challenged with IMQ and treated with PDRN (8 mg/kg/ip). In histological experiments an additional arm of IMQ animals was treated with PDRN plus istradefylline (KW6002; 25 mg/kg/ip, administered in vehicle consisting of a mixture of Tween and saline solution), an A2A antagonist. Both PDRN and istradefylline were daily administered starting on day 3, following psoriasis-like lesions induction. Mice from each group were killed 7 days following the first imiquimod or Vaseline administration.

### 4.2. Histology

Skin samples were fixed in 10% buffered formalin at room temperature for 24 h. Sections were then dehydrated in graded ethanol, cleared in xylene, and embedded in paraffin; five micrometer-thick sections of paraffin-embedded tissues were mounted on glass slides and stained with hematoxylin and eosin following hydration. Skin structure, erythema, scales, thickening, epidermal acanthosis, and inflammation were evaluated.

### 4.3. Western Blot Analysis

Skin samples were homogenized in lysis buffer (25 mM Tris/HCl, pH 7.4, 1.0 mM ethylene glycol tetraacetic acid, 1.0 mM ethylenediamine tetraacetic acid, 0.5 mM phenylmethyl sulfonylfluoride, 10 μg/mL aprotinin, 10 μg/mL leupeptin, 10 μg/mL pepstatin A, and 10 μL/mL NP40) and the lysates were centrifuged at 15.000 rpm for 15 min at 4 °C. Total protein content was measured using the Bio-Rad protein-assay kit (BioRad, Hercules, CA, USA) and the supernatant was mixed with Laemmli sample buffer (62 mmol/L Tris pH 6.8, 10% glycerol, 2% SDS, 5% β-mercaptoethanol, 0.003% bromophenol blue), and stored at −20 °C. Thirty micrograms of proteins were run by electrophoresis on a 10% sodium dodecyl sulphate (SDS) polyacrylamide gel. Following separation, proteins were transferred onto a PVDF membrane using a transfer buffer (39 mmol/L glycine, 48 mmol/L Tris, pH 8.3, 20% methanol) at 100 V for 1 h and then blocked with 5% non-fat dry milk in TBS 0.1% for 1 h at room temperature. Membranes were incubated with primary antibodies for TNF-α (1:500; Abcam, Cambridge, UK), pNF-κB p65 and β-actin (Cell Signaling, Danvers, MA, USA) in TBS-0.1% Tween overnight at 4 °C. Primary antibodies were removed, membranes were washed 3 times with TBS-0.1% Tween and incubated with a secondary peroxidase-conjugated goat anti-rabbit antibody (KPL, Gaithersburg, MD, USA) for 1 h at room temperature. Membranes were analyzed by the enhanced chemiluminescence system (LumiGlo reserve) according to the manufacture’s protocol (KPL) and the protein detection was quantified by scanning densitometry using a bio-image analysis system (C-DiGit, Li-cor, Lincoln, NE, USA). The results were expressed as relative integrated intensity compared to normal skin tissue measured within the same batch.

### 4.4. Real Time (RT) PCR Assay

Total RNA was extracted from skin of mice at the end of the experiment (7th day) and from human keratinocytes using Trizol LS reagent (Invitrogen, Carlsbad, CA, USA). RNA was quantified with a spectrophotometer (NanoDrop Lite, Thermo Fisher, Waltham, CA, USA) and 2 μg was reverse transcribed in a volume of 20 μL using the Superscript VILO kit (Invitrogen, Waltham, CA, USA). One microliter of cDNA was added to the EvaGreen qPCR Master Mix (Biotium Inc., Fremont, CA, USA), in a volume of 20 μL per well. Samples were run in duplicate and GADPH was used as housekeeping gene; the reaction was performed using the 2-step thermal protocol recommended by the manufacturer. Ten micromoles was the primer concentration selected to perform the analysis. Target genes were Wnt-1, β-catenin, IL-6, IL-10, IL-12, and IL-2.

Primers used for target and reference genes are listed below:

GADPH

Fw:5′GTCAAGGCTGAGAATGGGAA3′

Rv:5′ATACTCAGCACCAGCATCAC3′;

Wnt-1

Fw:5′ATAGCCTCCTCCACGAACCT3′

Rv:5′ GGAATTGCCACTTGCACTCT3′;

β-catenin

Fw:5′CGGCACCTTCCTATTTCTTCT3′

Rv:5′TCTGGAAATTAACTTCAGGCAAC3′;

IL-6

Fw:5′AGTTGCCTTCTTGGGACTGA3′

Rv:5′ TCCACGATTTCCCAGAGAAC3′;

IL-10

Fw:5′CCAAGCCTTATCGGAAATGA3′

Rv:5′ TTTTCACAGGGGAGAAATCG3′;

IL-12

Fw:5′CATCGATGAGCTGATGCAGT3′

Rv:5′CAGATAGCCCATCACCCTGT3′;

IL-2

Fw:5′CCCACTTCAAGCTCCACTTC3′

Rv:5′ATCCTGGGGAGTTTCAGGTT′;

Primers used for human target genes are listed below:

GAPDH

Fw:5′ GAGTCAACGGATTTGGTCGT3′

Rv:5′ TTGATTTTGGAGGGATCTCG3′;

IL6

Fw:5′ TACCCCCAGGAGAAGATTCC3′

Rv:5′ TTTTCTGCCAGTGCCTCTTT3′;

Wnt-1

Fw:5′ CGGCGTTTATCTTCGCTATC3′

Rv:5′ GCCTCGTTGTTGTGAAGGTT3′;

β-catenin

Fw:5′ATCTGCCTCCAGAGCAGGTA3′

Rv:5′CCTCAGGATTGCCTTTACCA3′;

Results were calculated using the 2-ΔΔC*t* method, and expressed as n-fold increase in gene expression using the CTRL group as calibrator.

### 4.5. Immunohistochemical Evaluation of CD3+ Cells

Paraffin-embedded tissues were sectioned (5 μm) and the immunohistochemical analysis was performed with a rabbit primary antibody anti-CD3 (ab18586, Abcam, Cambridge, UK) using the Ventana BenchMark automated slide stainer in combination with Ventana detection kits according to manufacturer specifications. Reaction was observed with diaminobenzidine tetra-hydrochloride (DAB; Sigma-Aldrich, Milan, Italy). Slides were counterstained with Mayer’s Haemalum, dehydrated, and mounted with coverslips. All slides were coded and evaluated by a pathologist at 40× magnification with a Nikon eclipse Ci-e microscope.

### 4.6. Cell Culture and Drug Treatment

Human HaCaT cells were obtained from ATCC (Manassas, USA) and cultured in high glucose Dulbecco’s modified Eagle’s medium (DMEM) with 10% inactivated FBS, 100-U/mL penicillin and 100-μg/mL streptomycin. Cells were incubated at 37 °C under 5% CO_2_. Starved HaCaT cells were stimulated with LPS (10 μg/mL) for 24 h or TNF-α (10 ng/mL) for 6h to reproduce an inflammatory condition in psoriatic keratinocytes. In addition, keratinocytes were treated with PDRN (1 μM), or CGS21680 (1 μM), or with PDRN in association with an A2A antagonist, ZM241385 (1 μM), for 4 h. ZM241385 was dissolved in DMSO to obtain a concentration of 1000 μM and then diluted in DMEM for a final concentration of 1 μM.

### 4.7. Statistical Analysis

All data are expressed as means ± standard deviation (S.D.). Different treatments were compared and analyzed by one-way ANOVA with Tukey post-test for intergroup comparisons. *P* < 0.05 was considered as the possibility of error and was considered as statistically significant. Graphs were drawn using GraphPad Prism (version 5.0 for Windows, GraphPad Software Inc, San Diego, CA, USA).

## Figures and Tables

**Figure 1 ijms-21-01215-f001:**
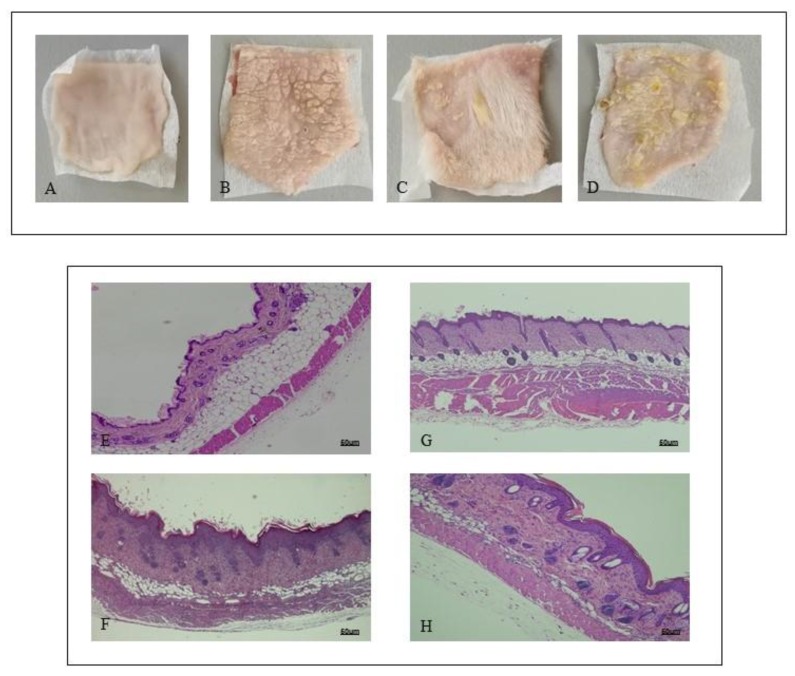
Representative macroscopic images of Sham (**A**), imiquimod (IMQ) (**B**), IMQ + PDRN (Polydeoxyribonucleotide) (**C**), IMQ + PDRN + KW6002 (**D**) mice back skin at day 7. Each image is representative of 10 animals per group. IMQ mice showed scales widely distributed (B) following imiquimod topical application; Sham mice skin appeared normal with any change following Vaseline application (A). Skin of mice treated with PDRN showed a significant reduction of scales (C). H&E stained sections of skin at day 7 observed under light microscopy (original magnification ×10) of Sham (**E**), IMQ (**F**) IMQ + PDRN (**G**) and IMQ + PDRN + KW6002 (**H**). Sham mice treated with Vaseline had a normal skin structure (E). IMQ application increased epidermal layers and caused acanthosis, papillomatosis and inflammatory infiltrate deposition (F). PDRN (8 mg/kg) reduced epidermal thickness, papillomatosis and inflammatory infiltrate (G) whereas KW6002 (25 mg/kg) administration abrogated PDRN effects (H). Scale bar: 50 µm.

**Figure 2 ijms-21-01215-f002:**
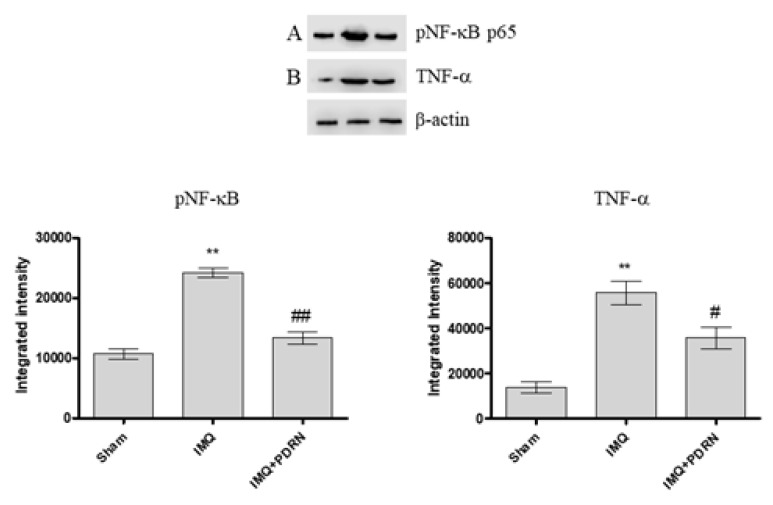
Western blot analysis of pNF-κB p65 (**A**) and Tumor Necrosis Factor alpha (TNF-α) (**B**) in skin samples of mice at day 7. IMQ animals showed a significant increase of both pNF-κB and TNF-α expression compared to Sham mice. Treatment with PDRN (8 mg/kg) significantly reduced the expression of proteins. Representative bands for each group have been reported and cropped from different gels for each antibody. Values were obtained from 10 animals per group and are expressed as mean and SD for each group. ** *p* < 0.001 vs. Sham; ## *p* < 0.001 and # *p* < 0.05 vs. IMQ.

**Figure 3 ijms-21-01215-f003:**
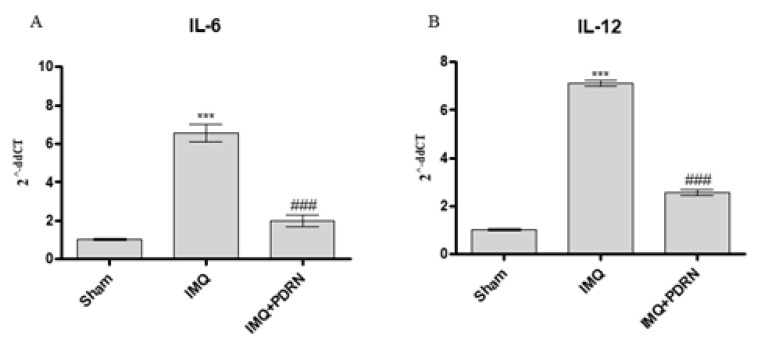
The graphs show qPCR results of Interleukins (IL)-6 (**A**) and IL-12 (**B**) mRNA expression in skin samples at day 7. IMQ application caused an increase of both IL-6 and IL-12 mRNA expression compared to Sham group. PDRN (8 mg/kg) reduced both mRNA expression. Values were obtained from 10 animals per group and are expressed as mean and SD for each group. *** *p* < 0.0001 vs. Sham; ### *p* < 0.0001 vs. IMQ.

**Figure 4 ijms-21-01215-f004:**
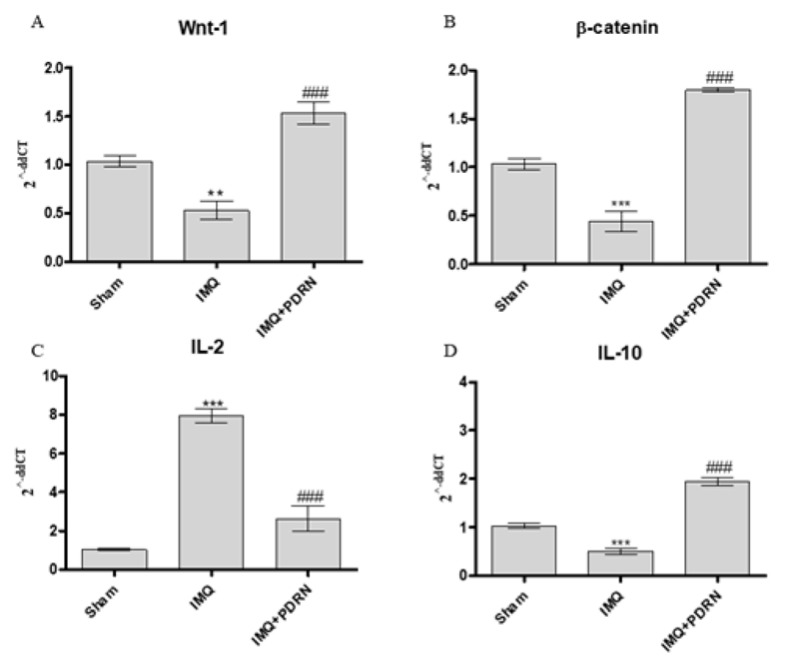
The graphs show qPCR results of Wnt-1 (**A**), β-catenin (**B**), IL-2 (**C**), and IL-10 (**D**) mRNA expression in skin samples at day 7. Mice challenged with IMQ showed a significant increase of Wnt-1, β-catenin, IL-2 and a significant decrease of IL-10 mRNA expression compared to Sham group. PDRN significantly reduced Wnt-1, β-catenin, IL-2 mRNA expression compared to IMQ group. On the contrary, PDRN treatment (8 mg/kg) caused a significant increase of IL-10 mRNA expression compared to IMQ mice. Values were obtained from 10 animals per group and are expressed as mean and SD for each group. ** *p* < 0.001 and *** *p* < 0.0001 vs. Sham; ### *p* < 0.0001 vs. IMQ.

**Figure 5 ijms-21-01215-f005:**
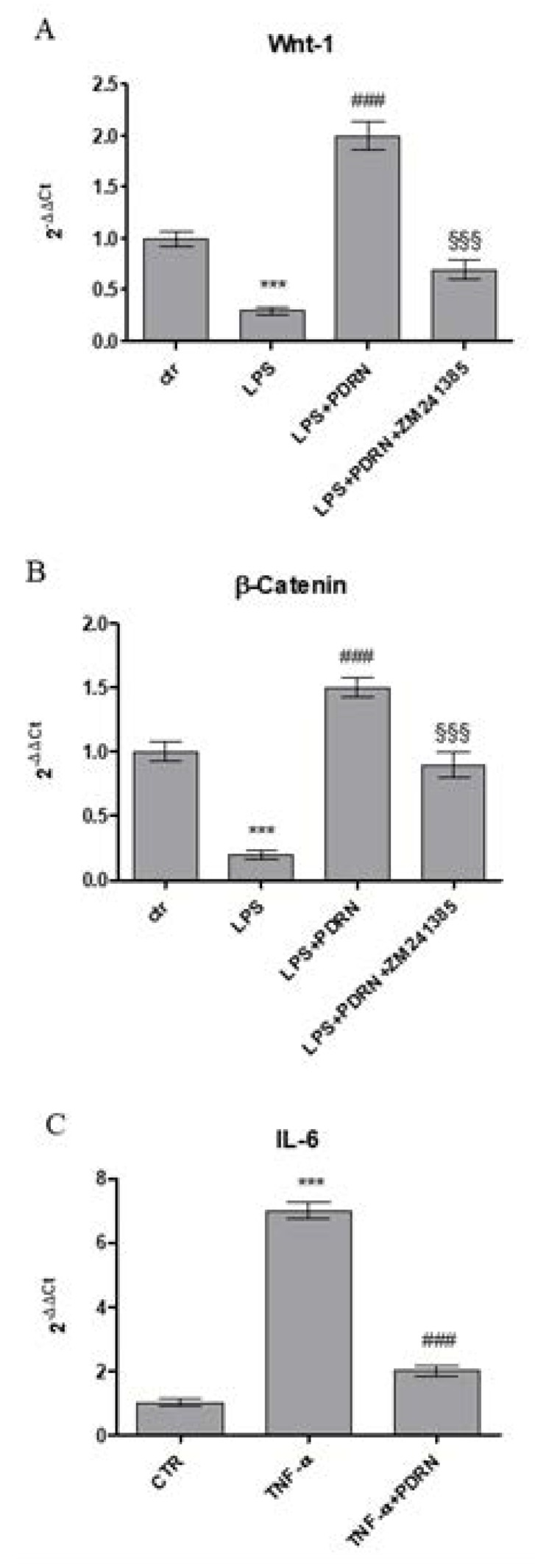
The graphs show qPCR results of Wnt-1 (**A**), β-catenin (**B**), and IL-6 (**C**) mRNA expression from human keratinocytes stimulated with LPS (10μg/mL) (**A**,**B**) treated with PDRN (1 μM) and with PDRN+ZM241385 (1 μM) or with TNF-α (10ng/mL) (**C**) treated with PDRN. Data are expressed as means ± SD. *** *p* < 0.0001 vs. Control (CTR); ### *p* < 0.0001 vs. LPS (A and B); §§§ *p* < 0.0001 vs. PDRN (A and B); ### *p* < 0.0001 vs. TNF-α (C).

**Figure 6 ijms-21-01215-f006:**
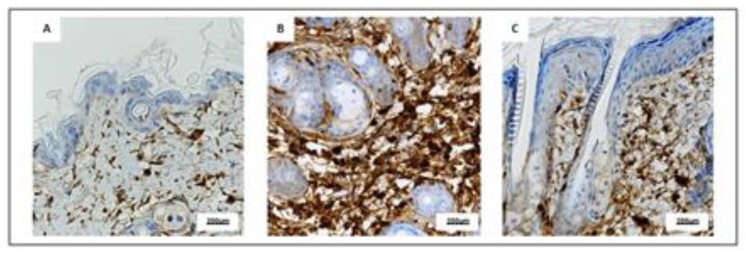
Representative immunostaining for CD3 (**A**–**C**) of skin sections at day 7 examined under light microscopy (original magnification X40) of Sham (**A**), IMQ (**B**), and IMQ+PDRN (**C**). IMQ animals showed an increased expression of CD3+ cells, whereas PDRN treatment (8 mg/kg) reduced CD3 immunostaining. Scale bar: 200 µm.

**Figure 7 ijms-21-01215-f007:**
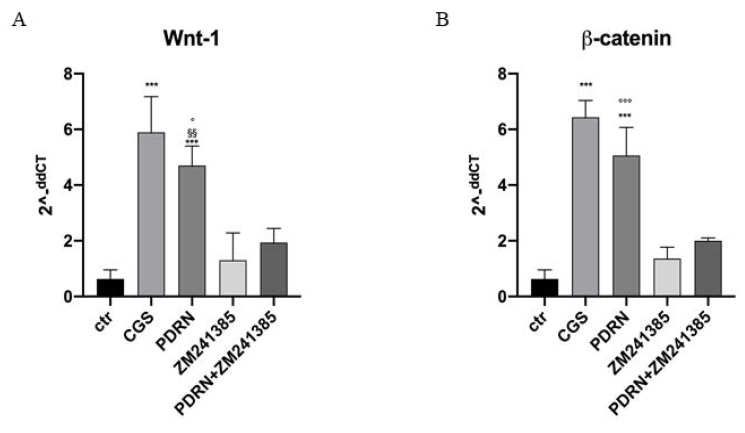
The graphs show qPCR results of Wnt-1 and β-catenin mRNA expression from human keratinocytes treated with PDRN (1 μM), CGS (1 μM), ZM241385 (1 μM) and PDRN+ZM241385 (1 μM). Data are expressed as means ± SD. *** *p* < 0.0001 vs. Control (CTR) and CGS vs. ZM241385; §§ *p* < 0.001 vs. ZM241385; ° *p* < 0.05 vs. PDRN+ZM241385 for the graph representing Wnt-1 expression; *** *p* < 0.0001 vs. CTR and ZM241385, °°° *p* < 0.0001 vs. PDRN+ZM241385 for the graph representing β-catenin expression.

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
