# Peer review of "PDRN, a Bioactive Natural Compound, Ameliorates Imiquimod-Induced Psoriasis through NF-κB Pathway Inhibition and Wnt/β-Catenin Signaling Modulation"

_ijms, 2020, doi:10.3390/ijms21041215_

Round 1
Reviewer 1 Report
In this work the authors show that treatment of animals with an A2A adenosine receptor agonist reduce inflammatory cytokines. These affects are associated with. Data have been obtained also in cells in vitro. The authors observed a reduced NFkB and enhanced WNT expression. The results are quite interesting but the conclusions would require further investigations to evaluate the molecular mechanisms through which A2A stimulation control inflammation in this model and in human samples.
The manuscript needs to be improved. Here the comments:
Page 2 line 18-20 the authors stated that “A2A receptor stimulation produces anti-inflammatory effects by inhibiting Nuclear factor κB (NF-κB) translocation to the nucleus and consequently suppressing TNF-expression. This could be relevant for the treatment of psoriasis.” Please add reference/s
Page 3 Figure 1 the quality of the images is poor. It is quite hard to note the histological features described in the text. Please improve the resolution. Please add scale bar. Images in Figure 1 are representative of how many animals per group? This should be clarified in the figure legend.
In figure 2 Western blotting results are shown. The authors should clarify which NF-kB subunit has been analysed. “Actina” should be corrected in “Actin”. In the legend of this figure the significance is reported: “**p<0.05 vs Sham; ##p<0.05 and #p<0.05 vs IMQ”. However, it is not clear why the authors stated that for all of them the p value is <0.05. Notably, this has been stated also in the legend of other figures including Fig 3, 4, 5 and 7.
In figure 6 scale bar should be added.
In material and methods section please specify animal treatment with PDRN. Is it daily administered?
Author Response
In agreement with the comments the following changes have been made throughout the manuscript and highlighted in yellow in the revised paper:
Page 2 line 18-20: As suggested by the Reviewer, we added a reference to support the statement.
Page 3 Figure 1: The Referee says that “… the quality of the images is poor…” In agreement with reviewer comment we improved the quality of the figures, we added a scale bar and we clarified the number of animals per group in the figure legend.
Figure 2: As requested by the Reviewer we clarified which NF-kB subunit has been analysed, we provided to change actina to actin and we checked the statistical significance between the several groups.
Figure 6: As suggested by the Reviewer we added the scale bar.
Material and Methods: As suggested by the Reviewer we specified that PDRN was daily administered.
Reviewer 2 Report
The objective of this study is to investigate the efficacy of polydeoxyribonucleotide (PDRN), an agonist of adenosine A2A receptor, in psoriasis. The authors demonstrated that PDRN treatment ameliorated skin alterations, decreased expression of pro-inflammatory cytokines, activated Wnt/β-catenin pathway, and reduced CD3+ cells in an experimental model of psoriasis-like dermatitis. Additionally, the authors showed that PDRN increased expression of Wnt-1 and β-catenin, which was abolished by an A2A receptor antagonist, in human keratinocytes. The authors conclude that PDRN inhibits nuclear factor-κB and modulates Wnt/β-catenin pathway through adenosine A2A receptor stimulation. There are some major concerns about the data that must be addressed.
Major comments:
The effects of the A2A receptor antagonist ZM241385 are hard to interpret in the absence of control experiments (Figure 7). The effect of ZM241385 without PDRN treatment should be presented. PDRN seems an agonist of the adenosine A2A receptor. To further confirm the involvement of A2A receptor in Wnt/β-catenin pathway, it would be worthwhile to show the effect of specific A2A receptor agonist on mRNA expression of Wnt-1 and β-catenin in human keratinocytes. The expression of adenosine receptors in skin are not presented. Ideally, immunostaining for A2A receptors of skin sections would be demonstrated.
Minor comments:
Page 11 lines 21–27, Please present the sequence of primers used for human housekeeping gene and IL6 in Materials and Methods.
Figure 1E–H and Figure 6, Please add scale bars.
Author Response
In agreement with the comments the following changes have been made throughout the manuscript and highlighted in yellow in the revised paper.:
Major comments
The Reviewer states: “The effects of the A2A receptor antagonist ZM241385 are hard to interpret in the absence of control experiments (Figure 7). The effect of ZM241385 without PDRN treatment should be presented. PDRN seems an agonist of the adenosine A2A receptor. To further confirm the involvement of A2A receptor in Wnt/β-catenin pathway, it would be worthwhile to show the effect of specific A2A receptor agonist on mRNA expression of Wnt-1 and β-catenin in human keratinocytes. The expression of adenosine receptors in skin are not presented. Ideally, immunostaining for A2A receptors of skin sections would be demonstrated”. In agreement with the Reviewer comments we performed additional experiments and we included new data in figure 7 regarding ZM241385 without PDRN treatment. In addition, we also presented additional experiments, as requested by the Reviewer, on the effect of a specific A2A receptor agonist on the Wnt-1/beta catenin signaling in human keratinocytes. Finally, we mentioned in the discussion section about a previous immunofluorescence experiment that demonstrated the positive expression of A2A receptors following PDRN treatment.
Minor comments
Page 11 lines 21-27: as requested by the Reviewer, we presented the sequence of primers used for human housekeeping gene and IL-6 in the Materials and Methods section of the revised manuscript.
Figure1 E-H and Figure 6: As requested by the Reviewer, we added scale bars
Reviewer 3 Report
Authors showed that PDRN (polydeoxyribonucleotide) inhibited inflammation in the animal model of psoriasis (in vivo and in vitro). The Nuclear factor NF-κB and Wnt/ ß-catenin signalling via A2A adenosine receptor is involved in the mechanism. PDRN reduced increased pNF-κB, TNF-α, of IL-6, IL-12, IL-2 induced by imiquimod. PDRN was also active in reversing increased by TNF-α IL-6 level and amount of CD3+ cells in skin. The effect of PDRN is suppressesed by adenosine A2A receptor antagonists, istradefilline and ZM241385 (in keratinocytes stimulated with LPS and TNF-α).
The work is very interesting and has translational potential, in particular the use of PDRN as topical formulation.
However, some issues need clarificatin. The information on affinity of PDRN to A2A receptors (also to other adenosine receptors) should be given, as agonist PDRN activity is mentioned in the work.
The effect of istradefilline and ZM241385 should be presented in figures.
The informtion how both antagonists were dissolved as it is known that both compounds have low water soulubility.
I suggest include doses/concentrations of compounds in figure legends as this information is mentioned only in the methods section.
Please, check the greek symbols (α, β) which are lacking in figure legends or the text body.
Author Response
In agreement with the comments the following changes have been made throughout the manuscript and highlighted in yellow in the revised paper.:
As requested by the Reviewer we commented on the affinity of PDRN to A2a receptors; we clarified how both antagonists were dissolved; we presented the effect of istradefylline and ZM241385 in figures; we included information on the doses/concentrations of compounds in figure legends; and we checked the Greek symbols in figure legends and text body.
Finally, we hope that now, after extensive revision and performing of new experiments, you will find our paper suitable for publication in the IJMS journal.
Round 2
Reviewer 1 Report
The revised manuscript is acceptable
Reviewer 2 Report
Manuscript has been sufficiently amended.